# Effects of Dietary Net Energy Concentration on Reproductive Performance, Immune Function, Milk Composition, and Gut Microbiota in Primiparous Lactating Sows

**DOI:** 10.3390/ani14203044

**Published:** 2024-10-21

**Authors:** Fang Gu, Lei Hou, Kaiguo Gao, Xiaolu Wen, Shuyun Mi, Guoxi Qin, Lijun Huang, Qiwen Wu, Xuefen Yang, Li Wang, Zongyong Jiang, Hao Xiao

**Affiliations:** 1Guangdong Key Laboratory of Animal Breeding and Nutrition, Ministry of Agriculture and Rural Affairs, Key Laboratory of Animal Nutrition and Feed Science in South China, State Key Laboratory of Swine and Poultry Breeding Industry, Institute of Animal Science, Guangdong Academy of Agricultural Sciences, Dafeng 1st Street, Guangzhou 510640, China; gf13633778817@163.com (F.G.); gaokaiguo@gdaas.cn (K.G.); wenxiaolu@gdaas.cn (X.W.); huanglijun0904@163.com (L.H.); wuqiwen@gdaas.cn (Q.W.); yangxuefen@gdaas.cn (X.Y.); wangli1@gdaas.cn (L.W.); 2Guangxi State Farms Yongxin Animal Husbandry Group Co., Ltd., No. 135 Qixing Road, Nanning 530022, China; rhoulei@126.com (L.H.); 15278959418@163.com (S.M.); qinguoxi2007@163.com (G.Q.)

**Keywords:** net energy, reproductive performance, immune function, milk composition, gut microbiota, primiparous lactating sows

## Abstract

**Simple Summary:**

Increased reproductive performance in modern high-yielding sows often results in inadequate energy intake. Many recent studies have focused on the improvement of sow energy intake through the addition of different dietary energy source, but there have few studies on dietary energy concentration based on the net energy system. Our results suggest that increasing the dietary energy concentration can increase immunological substances in milk and improve milk quality and body health in primiparous lactating sows.

**Abstract:**

This study aimed to determine the optimal dietary net energy concentration for the reproductive performance, immune function, milk composition, and gut microbiota of primiparous sows during lactation. Forty primiparous lactating sows (Landrace × Yorkshire) with similar body backfat thicknesses were randomly allocated into five treatment groups and fed diets with different dietary net energy concentrations of 10.05 MJ/kg, 10.47 MJ/kg, 10.89 MJ/kg, 11.30 MJ/kg, and 11.72 MJ/kg. The results showed that there were no differences in the performance of piglets, while there was a decrease in the daily feed intake of sows (*p* = 0.079, linear) as dietary net energy concentration increased. With the increasing dietary net energy concentration, the plasma insulin levels of sows increased (*p* < 0.01, linear), the plasma glucose levels tended to increase (*p* = 0.074, linear), and the blood urea nitrogen levels tended to decrease (*p* = 0.063, linear). Moreover, the plasma total superoxide dismutase activity of sows increased (*p* < 0.05, quadratic) and the plasma malondialdehyde content of sows decreased (*p* < 0.05, quadratic) by increasing the dietary net energy concentration. Interestingly, with the increase in dietary net energy concentration, the plasma immunoglobulin M content of sows increased, the milk immunoglobulin M, immunoglobulin G, immunoglobulin A and the percentage of milk fat increased (*p* < 0.05, linear), and the milk secretory immunoglobulin A content also increased (*p* < 0.05, linear and quadratic). The milk immunoglobulins and milk fat content of sows fed with net energy concentration of 11.72 MJ/kg were highest. Moreover, there were significant differences in the α-diversity, β-diversity, and relative abundance of gut microbiota in sows fed with different dietary net energy concentrations. At the phylum level, *Spirochaetota* and *Bacteroidota* in the gut microbiota of sows were mainly affected by increasing the dietary net energy concentration. Furthermore, the correlation analysis showed that milk immunoglobulin content had a significant negative correlation with the relative abundance of *Bacteroidota*, and plasma malondialdehyde content also had a significant negative correlation with the relative abundance of *Spirochaetota*. In summary, these results suggest that increasing the dietary net energy concentration to 11.72 MJ/kg can increase immunological substances in milk, improve milk quality, and alter the composition of gut microbiota in primiparous lactating sows.

## 1. Introduction

Milk secretion is the main energy-consuming behavior during lactation, accounting for 65% to 80% of the energy requirements of sows [1]. As the reproductive performance of modern high-yielding sows has improved, the energy requirements of lactating sows have also increased [2]. Studies have found that modern high-yielding sows do not normally obtain enough energy from the diet to fully satisfy their own needs [1,3,4]. Furthermore, sows often fail to reach their maximum feed intake capacity due to a variety of factors, including summer heat stress and feeding management issues. This leads to a lack of energy intake in lactating sows [5,6,7]. During lactation, when the energy intake cannot meet sows’ needs, sows will mobilize their own stored protein and fat to meet the need for milk secretion. However, excessive depletion of body-stored protein and fat can lead to the deterioration of sows’ body conditions, which in turn affects sows’ subsequent reproductive performance and may have permanent a negative impact on reproductive performance [8]. This effect is particularly severe for primiparous lactating sows [9].

Many previous studies have been conducted to improve the energy intake of sows by adding different dietary energy sources. Fats, as an important source of energy, not only augment the energy intake of sows but also influence their milk composition, thereby enhancing the growth performance of piglets [10]. Wang et al. analyzed 19 papers and found that the addition of fat to diets during late gestation and lactation led to a decrease in the daily feed intake but increased daily energy intake of lactating sows [11]. Several studies have shown that adding fat to the diet can increase milk production and milk fat content, which improves piglet performance [12,13]. However, there are few studies that focus on dietary net energy concentration. Xue et al. found that increasing the dietary energy concentration of lactation from 12.8 to 13.4 MJ ME/kg improved energy intake, consequently reduced the weight loss of sows, and increased the growth rate of piglets during lactation [14]. Rooney et al. also found that the diet of 15.2 MJ DE/kg significantly increased the energy intake of lactating sows and the litter weights of weaning piglets compared to dietary energy concentrations of 13.8 MJ DE/kg and 14.5 MJ DE/kg [15]. While current research on dietary energy concentration generally employs digestible or metabolized energy systems, the variation range of dietary energy concentrations is relatively small, which does not fully exploit the potential of dietary energy concentration [16]. Furthermore, the gut microbiota exerts a profound influence on the health and productivity of sows. It has been demonstrated that improving the gut microbiota of animals by modifying their diet has a beneficial impact on their overall health. Therefore, it is necessary to further investigate whether increasing the dietary net energy concentrations can improve the body health of lactating sows by affecting their gut microbiota. This study aimed to determine the optimal net energy concentration on the reproductive performance, immune function, milk composition, and gut microbiota of primiparous sows during lactation.

## 2. Materials and Methods

### 2.1. Animals and Experimental Design

Forty primiparous lactating sows (Landrace × Yorkshire) with similar backfat thicknesses (27.57 ± 0.47 mm) were randomly allocated into five treatment groups, with each group containing eight sows. All experimental sows were healthy and had been vaccinated in accordance with farm vaccination guidelines. In accordance with the NRC (2012) [17] recommended dietary net energy concentration of 10.47 MJ/kg, the net energy concentration of each treatment group was increased by 0.42 MJ/kg on the basis of the previous concentration from 10.05 MJ/kg. The dietary net energy concentrations of the treatment groups were 10.05 MJ/kg, 10.47 MJ/kg, 10.89 MJ/kg, 11.30 MJ/kg, and 11.72 MJ/kg. The birthday of the piglets was designated as day 0, and cross-fostering within the group was promptly carried out. The litter size was 12 for each sow after adjustment. The whole experimental cycle was 21 days, and if piglets died during this period, the weight of the dead piglet was measured and recorded.

### 2.2. Diets and Feeding

Corn–soybean meal diets (Table 1) were formulated according to the nutrient requirements of swine (NRC, 2012). Nutrient concentrations were kept consistent except for net energy concentration. Sows were kept in individual farrowing crates with the ad libitum feeding pattern and watered ad libitum. The experiment entailed the daily recording of feed intake, the leftover amount from the previous day, and the waste amount of the sows. This was performed in order to calculate sows’ actual daily feed intake.

### 2.3. Sample Collection

On the 14th day of lactation, the blood samples were collected from sows after feeding for 2 h and centrifuged at 1000× *g* for 10 min at 4 °C to separate plasma samples, which were then stored at −80 °C. Milk samples were collected on the 14th day of lactation. Each sow was injected with 2 mL of oxytocin (1 mL:10 IU) at the ear margin, and equal amounts of milk samples were collected from the anterior, middle, and posterior teats using a massage and squeezing technique. From each sow, 40 mL milk samples were collected, divided and labeled, and then stored at −80 °C. On the 21st day of lactation, fresh fecal samples were collected from sows using sterile test tubes and stored at −80 °C.

### 2.4. Performance Measurement of Sows and Piglets

The thickness of backfat in sows was measured using an ultrasonic device (RENCO, Manchester-by-the-Sea, MA, USA) at the P2 position (6 cm from the midline at the head of the last rib) on days 0 and 21 of lactation. The loss of backfat during lactation was calculated. The per piglet weight was recorded on day 0, and the piglets were weighed after being fasted for 12 h on day 21. The average daily gain (ADG) was obtained by subtracting the average piglet weight per litter at the beginning from the average piglet weight per litter at the end and dividing by the number of nursing days.

### 2.5. Chemical Analyses

The milk samples were centrifuged at 30,000× *g* for 20 min at 4 °C to separate whey samples. Several indicators in the whey samples were analyzed using enzyme-linked immunosorbent assay (ELISA) kits (Jiangsu Meimian Industrial Co., Ltd., Yancheng, China), including the level of immunoglobulin A (MM-0905O1), secretory immunoglobulin A (MM-36234O1), immunoglobulin M (MM-0402O1), and immunoglobulin G (MM-0403O1) in plasma and milk on day 14 of lactation. The percentage of protein, lactose, fat, and nonfat solids in milk was measured using a milk composition analyzer (EKOMILK, Stara Zagora, Bulgaria). The content of glucose (F006-1-1) and blood urea nitrogen (C013-2-1) in plasma was measured with commercial kits according to the manufacturer’s instructions (Nanjing Jiancheng Biotechnology Co. Ltd., Nanjing, China). The contents of progesterone (MM-1205O1), estradiol (MM-0480O1), and insulin (MM-0390O1) were measured in plasma using ELISA kits (Jiangsu Meimian Industrial Co., Ltd., Yancheng, China). Total superoxide dismutase (A001-1-2) activity, total antioxidant capacity (A015-2-1), and malondialdehyde (A003-1-2) content in plasma were determined using a kit (Nanjing Jiancheng Biotechnology Co. Ltd., Nanjing, China) according to the manufacturer’s instructions. Before detecting total superoxide dismutase in the plasma sample, we diluted the plasma fourfold with 0.9% normal saline.

### 2.6. Microbiota Profiling

As described above, total microbiome DNA was extracted from sow fecal samples according to the E.Z.N.A.^®^ Soil DNA Kit (Omega Bio-tek, Norcross, GA, USA) instructions [18]. 16s rRNA V4-V5 region primers were used for PCR amplification. PCR amplification experiments were performed as follows: 4 min at 94 °C; 30 s at 94 °C, 30 s at 55 °C, and 1 min at 72 °C. The above three steps were performed for 25 cycles and finally held at 72 °C for 10 min. The amplified products were purified with 2% agarose gel electrophoresis using an AxyPrep DNA Gel Extraction Kit (Axygen Biosciences, Union City, CA, USA) according to the instruction procedure. The purified PCR products were accurately quantified using Qubit^®^ 3.0 (Thermo Fisher Scientific, Waltham, MA, USA), and then 24 amplicon samples with different tag sequences were mixed in equal amounts. The mixed pooled DNA products were used to construct a PE library for Illumina double-end sequencing by following the Illumina genome sequencing library construction process. The amplicon library was sequenced in PE250 mode using the Illumina platform (Shanghai BIOZERON Co., Ltd., Shanghai, China) according to the standard procedure. The raw data were obtained as valid data after high-quality QC and chimera removal. All sequences in the valid data were categorized according to different similarity levels to obtain operational taxonomic units (OTUs). Chimeric sequences were identified and removed using UCHIME 4.2.40 software, and sequences with 97% similarity were clustered into OTUs using UPARSE (version 7.1) software. The RDP classifier Bayesian algorithm was used for the taxonomic analysis of the OTU representative sequences compared to the Silva database, with a confidence threshold of 0.7, to finally obtain species information for each OTU at each taxonomic level, and from this, the microbial community composition of each sample at each taxonomic level was counted. The complexity of species diversity was evaluated using ACE, Shannon, Simpson, and Chao. The β-diversity analysis was performed using Bray–Curtis to compare the results of the principal coordinates analysis (PCoA) and non-metric multidimensional scaling (NMDS). The relative abundance of major microorganisms (except unidentified species) at the phylum to genus level was analyzed to detect whether the composition of the microbial community was altered by increasing dietary net energy concentration.

### 2.7. Statistical Analysis

All data results were presented as the mean and standard error (SEM). Statistical analyses were conducted using the one-way variance analysis (ANOVA) of the statistical software SPSS 25.0 based on the dietary net energy concentration model, with each individual sow as an experimental unit. Dietary net energy concentration was specified as a fixed effect. In the model, the response variable included the average daily feed intake of sows, backfat thickness, the average daily gain of piglets, plasma biochemical indices, plasma hormone levels, plasma antioxidant capacity, milk composition, and gut microbiota. Multiple comparisons were made when the ANOVA indicated significant differences. Tukey’s test was used for multiple comparisons. Orthogonal polynomial contrasts were performed to evaluate the linear and quadratic effects of the dietary net energy concentration. Statistical significance was declared at *p* < 0.05, tendencies were declared at 0.05 < *p* < 0.10, and high statistical significance was declared at *p* < 0.05.

## 3. Results

### 3.1. Performance of Sows and Piglets

The effects of dietary net energy concentration on the performance of sows and piglets are shown in Table 2. Increasing the dietary net energy concentration tended to decrease the daily feed intake of sows linearly (*p* = 0.079). There were no differences in the daily net energy intake, backfat thickness, backfat loss during lactation of sows, and the growth performance of piglets among the five treatment groups (*p* > 0.10).

### 3.2. Plasma Biochemical Indices and Hormone Levels

As shown in Table 3, there was a linear effect of increasing the dietary net energy concentration on the plasma INS levels of sows (*p* < 0.01). Increasing the dietary net energy concentration tended to increase the plasma GLU levels of sows linearly (*p* = 0.074) and decrease the BUN levels of sows linearly (*p* = 0.063). There was no effect of different dietary net energy concentrations on plasma E2 and PROG levels of sows among the five groups (*p* > 0.10).

### 3.3. Plasma Antioxidant Capacity

The effects of dietary net energy concentration on the antioxidant ability of sows are shown in Table 4. Increasing the dietary net energy concentration increased the plasma T-SOD activity of sows quadratically (*p* < 0.01) and decreased the MDA content quadratically (*p* < 0.05). Moreover, the plasma MDA content of sows fed diets with a net energy concentration of 10.47 MJ/kg was the highest, and the plasma T-SOD activity was the lowest. There were no significant differences in the plasma T-AOC among groups (*p* > 0.10).

### 3.4. Milk Composition

The effects of dietary net energy concentration on the composition of milk are shown in Table 5. With increasing dietary net energy concentration, the percentage of milk fat increased linearly (*p* < 0.01). There was no significant effect of different dietary net energy concentrations on lactoprotein, lactose, and nonfat solids among the five groups (*p* > 0.10).

### 3.5. Immunoglobulins

The effects of dietary net energy concentration on plasma and milk immunoglobulins are summarized in Table 6. With increasing dietary net energy concentration, the content of IgM in plasma and the contents of IgM, IgG, IgA, and SIgA in the milk of sows increased linearly (*p* < 0.05). With increasing dietary net energy concentration, the content of SIgA in the milk of sows also increased quadratically (*p* < 0.05). There was no effect of different net energy concentrations on plasma IgA and IgG content among the five groups (*p* > 0.10).

### 3.6. Gut Microbiota

The effects of dietary net energy concentration on the gut microbiota of sows are shown in Figure 1. As shown in Figure 1A, the Chao index was significantly increased in the 10.89 MJ/kg group compared to the 10.47 and 11.30 MJ/kg group (*p* < 0.05). The Shannon index of sows fed diets with a net energy concentration of 11.30 MJ/kg was lower than the 10.47 and 10.89 MJ/kg group (*p* < 0.05). The Simpson index of sows fed diets with a net energy concentration of 11.30 MJ/kg was significantly higher than the 10.47, 10.89, and 11.72 MJ/kg group (*p* < 0.05). The PCoA and NMDS plots were employed to assess the differences in β-diversity. As shown in Figure 1B,C, the comparison of the distribution of community structure between the 10.05 MJ/kg group and the 11.30 MJ/kg group revealed differences. Moreover, there were differences in the distribution of community structure between the 10.05 MJ/kg group and the 11.72 MJ/kg group.

The effects of dietary net energy concentration on the relative abundance of various gut microbiota of sows are shown in Figure 2A. As can be seen in Figure 2B, the relative abundance of specific gut microbiota significantly increased in the 10.89 MJ/kg group compared to the 10.05 MJ/kg group (*p* < 0.05), such as *Spirochaetota* at the phylum level, *Spirochaetia* at the class level, *Spirochaetales* at the order level, *Spirochaetaceae* at the family level, and *Treponema* at the genus level. Increasing the dietary net energy concentration decreased the relative abundance of certain gut microbiota (*p* < 0.05). These included *Bacteroidota* at the phylum level, *Bacteroidia* at the class level, and *Bacteroidales* at the order level. Moreover, as shown in Figure 2C, the ratio of *Firmicutes* to *Bacteroidota* increased significantly as dietary net energy concentration increased (*p* < 0.05). As shown in Figure 2D, the dominant gut microbial genera in the 10.05 MJ/kg group were *Eubacterium_siraeum_group*, *Cellulosilyticum*, and *Lachnoclostridium*; in the 10.47 MJ/kg group, the dominant gut microbial genera were *Monoglobus*, *Prevotella_7*, and *Prevotella*; in the 10.89 MJ/kg group, the dominant gut microbial genera were *Prevotella_9*, *Clostridium_sensu_stricto_6*, and *Prevotellaceae_NK3B31_group*; in the 11.30 MJ/kg group, the dominant gut microbial genus was *Shttleworthia*; and in the 11.72 MJ/kg group, the dominant gut microbial genus was *Streptococcus*.

### 3.7. Spearman’s Correlation Analysis

The results of the Spearman’s correlation analysis of the gut microbiota with milk composition, milk immunoglobulins, plasma antioxidant ability, plasma hormone levels, and biochemical indices are shown in Figure 3. *Spirochaetota* (phylum), *Spirochaetia* (class), *Spirochaetales* (order), *Spirochaetaceae* (family), and *Treponema* (genus) were negatively correlated with plasma MDA content in sows (*p* < 0.05). *Bacteroidota* (phylum), *Bacteroidia* (class), and *Bacteroidales* (order) were negatively correlated with the SIgA, IgA, IgG, and IgM contents of milk. (*p* < 0.05).

## 4. Discussion

Dietary energy concentration and feed intake are important determinants of energy intake in lactating sows. A previous study has shown that increasing the dietary net energy concentration improved the energy intake and performance of sows and the growth of suckling piglets [14,19]. In this experiment, there was a tendency for the daily feed intake of sows to decrease as dietary net energy concentration increased. This finding is in accordance with the study conducted by Lu et al. [20]. Their study found that increasing dietary net energy concentration from 8.83 MJ/kg to 11.43 MJ/kg resulted in a linear decrease in the average daily feed intake of barrows and gilts. Interestingly, in their study, compared to pigs fed an 11.43 MJ NE/kg diet, even though those fed an 8.83 MJ NE/kg diet had an 8.7% increment on ADFI, the daily net energy intake was still 5.4% less. This is consistent with our results. In our experiment, although there was no significant difference in energy intake, the daily net energy intake of sows in the 11.72 MJ/kg group increased by 5.73 MJ compared with the sows in the lowest energy group (10.05 MJ/kg). The amount of food intake during lactation is affected by multiple factors, such as the genetic background of sows [21], ambient temperature [22,23], delivery body condition, parity, etc.; in particular, the high-temperature environment in summer will limit the food intake of sows. In this trial, increasing the dietary net energy concentration had no significant effect on the growth performance of suckling piglets. Although it has been shown that increasing the energy concentration of sows’ diets improves piglets’ growth performance, this phenomenon was generally shown in the trial with multiparous sows, while piglets’ growth performance was not affected in the trial with primiparous sows [12,14]. This should be due to the differences in the physiological metabolism between primiparous sows and multiparous sows. The body development of primiparous sows is not yet complete, and the first intense lactation may lead to an increase in the catabolism of body tissue. When their energy intake is insufficient, sows will break down their body tissue to meet lactating nutrient needs. In addition, primiparous sows have lower feed intake than multiparous sows [14]. Rooney et al.’s report is consistent with this [15]. Changing the dietary net energy concentration did not significantly affect the performance of multiparous sows. However, they found that increasing the dietary energy concentration of sows from 14.5 MJ DE/kg to 15.9 MJ DE/kg increased the average daily gain of suckling piglets by 21 g/d. In this study, it was noted that an increase in net energy intake of 5.73 MJ per day did not have a significant effect on backfat loss in primiparous sows. This may be because body fat reserves are the most backward in the energy allocation order of lactating sows, and the increase of 5.73 MJ NE in the daily energy intake cannot affect the body storage of sows. Previous studies also found that an increase of 7.91 MJ ME in daily energy intake had no significant effect on weight loss in primiparous sows [14].

Milk production is the main energy output behavior of lactating sows. Milk composition is closely related to the nutritional intake of lactating sows. In a previous study, Pedersen et al. found that the change in dietary energy concentration had a relatively weak effect on the milk production of sows [13]. Therefore, we examined the composition of milk. In this experiment, increasing dietary net energy concentration increased the percentage of milk fat very significantly. This is consistent with the findings of Lellis and Speer [24]. Milk fat plays an important role in regulating the body temperature of piglets and affects their early survival rate and body fat percentage at weaning [25]. The observed increase in milk fat in this trial is likely to be related to the choice of soybean oil as an energy source. A lot of studies have indicated that the fatty acids in milk are derived from blood lipids (both endogenous and dietary fatty acids) [26,27]. Fatty acids may preferentially shunt directly to the mammary gland, thereby increasing the raw material available for mammary lipid synthesis. Consequently, the addition of fat to the diet will result in an increase in the percentage of milk fat [12,28]. In contrast, a high-starch diet may lead to a significant increase in sows’ lactose content. In addition to milk fat, acquired immunity provided by immunoglobulins in milk is an important factor in piglets’ health [29,30]. Increasing the concentration of immunoglobulins in milk improved the immune function of piglets [31]. Since piglets are susceptible to stress from external stimuli, it is necessary to increase the level of immunoglobulins ingested to improve their health [32]. The observed increase in milk immunoglobulins may be attributed to the fact that an adequate energy supply is conducive to the optimal functioning of the immune system, which in turn stimulates the synthesis of immunoglobulins [33]. In the present study, it was found that the contents of IgG, IgM, IgA, and SIgA in milk increased linearly as the dietary net energy concentration increased. We observed that the daily weight gain of piglets did not show a significant increase in this experiment. This may indicate that milk fat and immunoglobulins primarily improve piglet health, survival rate, and subsequent growth performance, rather than immediate growth performance [25,34,35]. In addition, there are few studies on the effect of net energy concentrations on immunoglobulin in sow milk. The specific reason needs further exploration.

A previous study found that increasing energy intake from 33.49 to 66.99 MJ ME resulted in a decrease in serum urea levels in sows, independent of protein intake [36]. In the present study, BUN levels were reduced in groups with high dietary net energy concentrations, which is in agreement with Nelssen’s findings. This may imply that the high energy intake mitigates the consumption of self-body protein by sows. Insulin is a key hormone in the regulation of energy metabolism. Long-term high energy intake will lead to insulin resistance, which will cause inflammation and disease in the body [37,38]. Therefore, there is a need to assess the risks of high-energy diets. In this study, plasma INS levels in sows increased with increasing dietary net energy concentration, whereas the plasma GLU level also demonstrated a tendency to increase linearly with increasing net energy concentrations. Although sows with hyperglycemia did not demonstrate a significant increase in backfat, their feed intake was affected to some extent. This may be related to insulin resistance, which requires further investigation [39]. Interestingly, in this trial, although the daily energy intake of sows was not statistically different among dietary treatments, there were significant changes in plasma insulin and GLU levels. This may be associated with the types of energy feedstocks employed. In this experiment, we increased the dietary net energy concentration by adding soybean oil. Although energy intake was generally consistent, hyperglycemic sows actually ate more fat. It has been demonstrated that a diet high in fat may result in transient insulin resistance [40]. This transient insulin resistance may result in elevated blood glucose levels, particularly when carbohydrates are consumed concurrently [41].

Studies have found that increasing the metabolic load in lactating sows leads to elevated systemic oxidative stress [42]. Excessive free radical production may lead to lipid and protein oxidation and impaired normal endothelial cell function [43]. In the present study, increasing the dietary energy concentration decreased the plasma MDA content and increased the plasma T-SOD activity in sows. This may be related to the content of soybean oil in diets. Studies have shown that vitamin E, phytosterol, and phospholipids in soybean oil have certain antioxidant effects [44,45,46]. Interestingly, the highest plasma MDA content and the lowest plasma T-SOD activity were observed when the dietary net energy concentration was 10.47 MJ/kg, but not when the dietary net energy concentration was 10.05 MJ/kg. It is possible that the dietary net energy concentration of 10.05 MJ/kg contained a greater proportion of soybean hulls. Soybean hulls are rich in isoflavones, which have a better antioxidant effect than soybean oil [47]. Our results indicated that adding soybean oil increased the dietary net energy concentration but did not increase the risk of oxidative stress in sows. Instead, it appears to exert a beneficial effect on the sows’ antioxidant capacity.

The gut microbiota of sows plays a vital role for their hosts, helping them to digest nutrients, provide vitamins and beneficial metabolites, and improve host production performance. Furthermore, gut microbiota exerts a modulating effect on the body’s immunity [48]. In the present study, sow milk immunoglobulins exhibited a positive correlation with increasing dietary net energy concentration. Therefore, we conducted an analysis of the gut microbiota of sows with the objective of investigating the association between them and sow immunoglobulins. Lu et al. reported that as pigs age, their microbiota composition becomes less dependent on the environment [49]. The gut microbiota of lactating sows is mature, stable, and less affected by external influences. However, the diversity and relative abundance of gut microbiota in sows in the high-energy group yielded significant differences compared to the low-energy group. *Firmicutes*, *Bacteroidota*, and *Spirocchaetota* were the most dominant phyla in all samples in this study, which is in agreement with the report by Wang et al. [50]. An increase in the relative abundance of *Firmicutes* is usually associated with high energy intake from the diet, whereas an increase in the relative abundance of *Bacteroidota* is associated with weight loss [51]. Therefore, a high ratio of *Firmicutes*/*Bacteroidota* is expected to be used to improve the growth performance of pigs. In the present study, high-energy diets did not significantly affect the relative abundance of *Firmicutes* in sows, but increasing the dietary net energy concentration decreased the relative abundance of *Bacteroidota* and increased the *Firmicutes*/*Bacteroidota* ratio. Correlation analyses showed that *Bacteroidota* was negatively correlated with sow milk immunoglobulin content.

*Treponema* is considered to be part of the so-called “core microbiota” of healthy pigs [52]. Previous studies have found that the relative abundance of *Treponema* correlates with average daily weight gain and increased backfat in pigs [53]. This microbiota is primarily involved in the digestion of dietary *polysaccharides* (fiber, including cellulose and lignin), as well as the production of large amounts of short-chain fatty acids [54]. *Treponema* was found to be positively correlated with the apparent digestibility of crude fiber by Niu et al. [55]. Interestingly, the experimental group with high fiber content in this study did not show high *Treponema* relative abundance, which may be related to the fact that the variation of *Treponema* relative abundance is affected by a variety of reasons, such as host genetic background, dietary fiber, and bacterial competition. In addition, from the correlation analysis, *Treponema* was negatively correlated with the plasma MDA content of sows, which had a certain degree of influence on the antioxidant capacity of the sow organism. Therefore, the effect of dietary energy concentration on *Treponema* and its metabolites needs to be further investigated.

Although there was a correlation between the differential microbiota in this trial and sow immunoglobulin, no significant immunological effect was found for the differential microbiota itself. It has been demonstrated that the activity of immune cells is dependent on the supply of energy [56]. Insufficient energy will result in a reduction in the proliferation, differentiation, and effector functions of immune cells. Consequently, the observed increase in milk immunoglobulins in sows in this experiment is more likely to be attributable to an increased supply of energy to immune cells rather than to the effects mediated through intestinal microorganisms. In addition, this study investigated the effects of different dietary net energy concentrations on the lactation performance of primiparous sows. In the actual production, we can adjust the diet formula according to the results and then improve the production efficiency.

## 5. Conclusions

In summary, these results suggest that increasing the dietary net energy concentration to 11.72 MJ/kg can increase immunological substances in milk, improve milk quality, and alter the composition of the gut microbiota of primiparous lactating sows.

## Figures and Tables

**Figure 1 animals-14-03044-f001:**
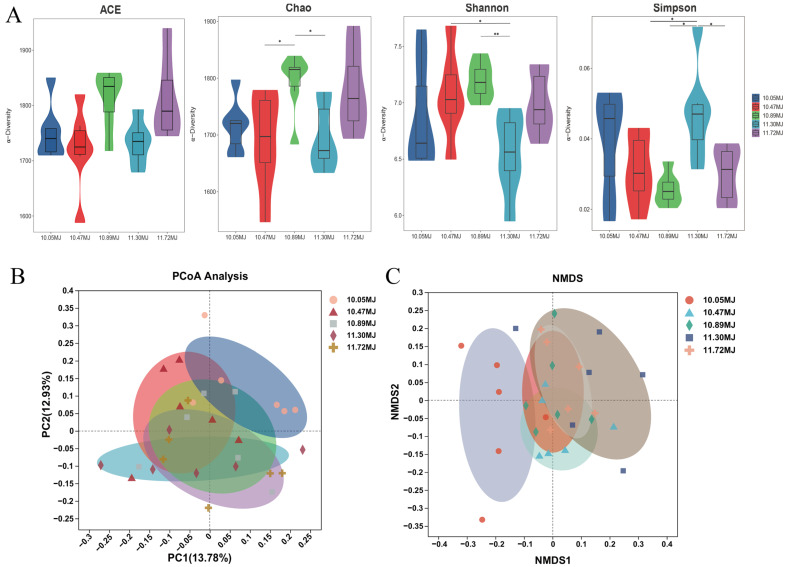
Effects of different dietary net energy concentrations on α-diversity and β-diversity of sows. (**A**) Results of alpha diversity. (**B**) Principal co-ordinates analysis (PCoA) plot visualizing among sample beta diversity of microbiota. (**C**) Non-metric multidimensional scaling (NMDS) plot visualizing among sample beta diversity of microbiota. Values are presented as means ± SEM, *n* = 6. Differences were assessed using Tukey’s test for multiple comparisons and denoted as follows: * means a significant difference (*p* < 0.05) and ** means a highly significant difference (*p* < 0.01).

**Figure 2 animals-14-03044-f002:**
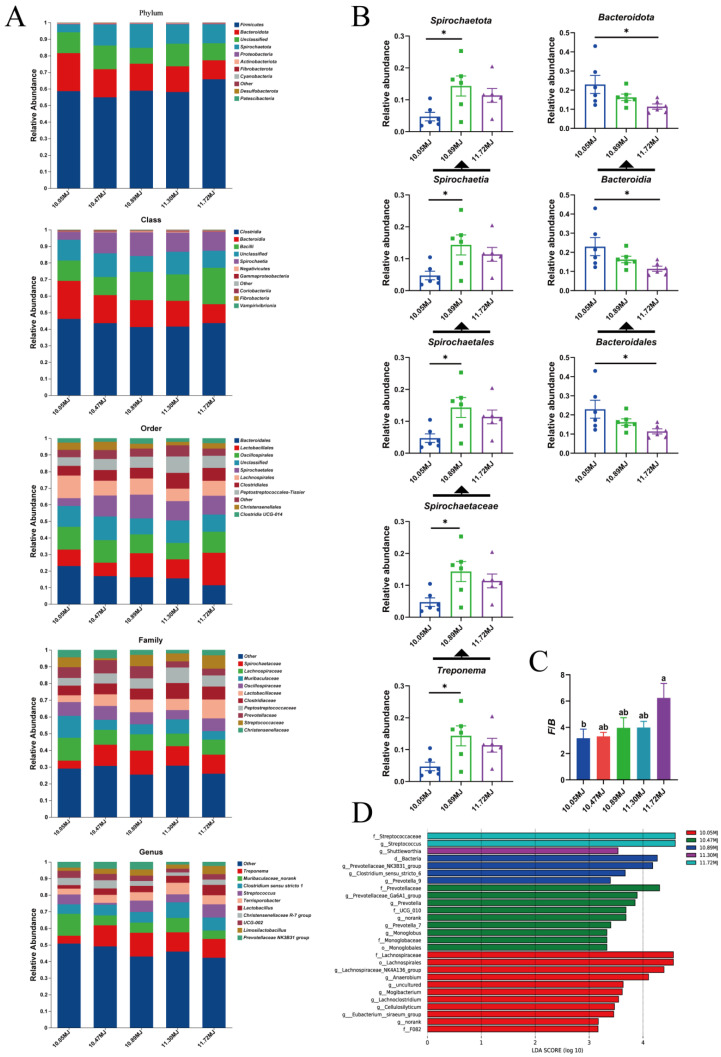
Gut microbial composition of sows in different treatment groups. (**A**) Relative abundances of main taxa at different levels. (**B**) Relative abundances of differential microflora at different levels. (**C**) Ratio of *Firmicutes* to *Bacteroidota* of sows in different treatment groups. (**D**) Differences in abundance, returned by LEfSe analysis. Values are presented as means ± SEM, *n* = 6. Differences were assessed using Tukey’s test for multiple comparisons and denoted as follows: * means a significant difference (*p* < 0.05). ^a,b^ means without a common letter indicate a significant difference (*p* < 0.05).

**Figure 3 animals-14-03044-f003:**
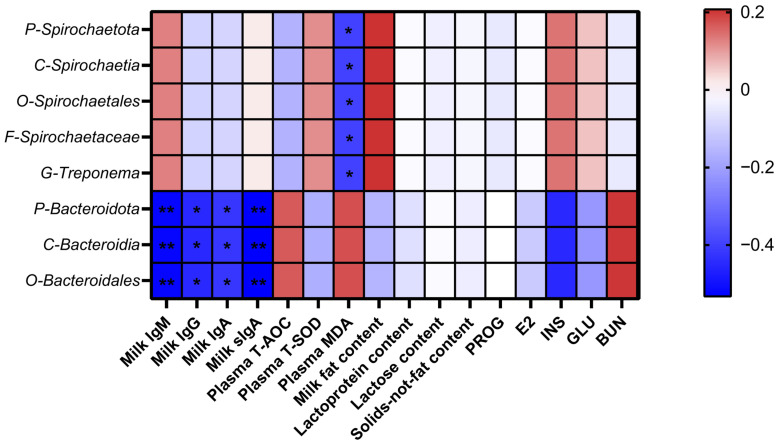
Spearman’s correlation analysis of significantly differential gut microbiota with milk composition, milk immunoglobulins, plasma antioxidant levels, plasma hormone levels, and biochemical indices in sows (P- = phylum; C- = class; O- = order; F- = family; G- = genus). * means a significant correlation (*p* < 0.05) and ** means a highly significant correlation (*p* < 0.01).

**Table 1 animals-14-03044-t001:** Composition and nutrient level of the diets for lactating sows (air-dry basis).

Item	Dietary Net Energy Concentration, MJ/kg				
	10.05	10.47	10.89	11.30	11.72
Ingredient, %					
Corn, 8%	58.80	58.20	57.86	57.34	56.74
Wheat meal	5.00	5.00	5.00	5.00	5.00
Soybean meal, 44%	21.50	22.18	22.60	23.19	23.85
Fish meal, 63%	3.00	3.00	3.00	3.00	3.00
Soybean hulls	8.00	6.00	4.00	2.00	0
Soybean oil	0	1.90	3.80	5.70	7.60
Choline chloride, 50%	0.25	0.25	0.25	0.25	0.25
Salt	0.45	0.45	0.45	0.45	0.45
CaHPO4	1.10	1.10	1.10	1.10	1.10
L-Lysine-HCl	0.10	0.10	0.10	0.10	0.10
Limestone	0.80	0.82	0.84	0.87	0.91
Premix ^a^	1.00	1.00	1.00	1.00	1.00
Total	100.00	100.00	100.00	100.00	100.00
Calculated composition ^b^					
ME, MJ/kg	13.51	13.97	14.43	14.89	15.35
NE, MJ/kg	10.05	10.47	10.89	11.30	11.72
CP, %	17.8	17.8	17.8	17.8	17.8
SID Lysine, %	1.07	1.07	1.07	1.07	1.07
SID Methionine + Cysteine, %	0.63	0.63	0.63	0.63	0.63
SID Threonine, %	0.71	0.71	0.71	0.71	0.71
SID Tryptophan, %	0.22	0.22	0.22	0.22	0.22
Ca, %	0.8	0.8	0.8	0.8	0.8
P, %	0.6	0.6	0.6	0.6	0.6

^a^ The premix provided the following per kg of the diet: vitamin A 10,000 IU, vitamin D 1400 IU, vitamin E 44 mg, vitamin K 3.0 mg, vitamin B 10.50 mg, vitamin B12 0.04 mg, nicotinic acid 45 mg, pantothenic acid 20 mg, folic acid 1.2 mg, biotin 0.20 mg, choline chloride 550 mg, Cu 80 mg, Fe 100 mg, Zn 100 mg, Mn 50 mg, I 0.3 mg, and Se 0.25 mg. ^b^ Calculated chemical concentrations using nutritional values for feed ingredients from the NRC (2012).

**Table 2 animals-14-03044-t002:** Effects of different dietary net energy concentrations on performance of sows and piglets.

Item	Dietary Net Energy Concentration, MJ/kg					SEM	*p*-Value		
	10.05	10.47	10.89	11.30	11.72		ANOVA	Linear	Quadratic
Daily feed intake, kg/d									
Lactation d1 to 21	5.18	5.15	4.98	4.30	4.93	0.11	0.082	0.079	0.358
Energy intake, MJ/d									
Lactation d1 to 21	52.09	53.92	54.27	48.59	57.82	1.20	0.172	0.455	0.366
Sow backfat thickness, mm									
Lactation d0	26.50	27.50	26.83	28.50	28.50	0.47	0.561	0.149	0.934
Lactation d21	21.50	23.67	21.67	21.83	24.67	0.49	0.145	0.181	0.374
Backfat loss d0–21	−5.00	−3.83	−5.17	−6.67	−3.83	0.42	0.191	0.863	0.361
Average weight of piglets, kg									
Lactation d0	1.71	1.70	1.58	1.53	1.54	0.05	0.657	0.158	0.789
Lactation d21	6.65	6.46	6.45	6.13	6.56	0.15	0.878	0.661	0.500
ADG of piglets	0.24	0.23	0.23	0.22	0.24	0.05	0.866	0.984	0.472

SEM standard error of the mean.

**Table 3 animals-14-03044-t003:** Effects of different dietary net energy concentrations on plasma biochemical indexes and hormone levels in lactating sows.

Item	Dietary Net Energy Concentration, MJ/kg					SEM	*p*-Value		
	10.05	10.47	10.89	11.30	11.72		ANOVA	Linear	Quadratic
GLU, mmol/L	4.48	4.79	4.27	5.35	5.18	0.16	0.149	0.074	0.606
BUN, mmol/L	12.34 ^ab^	12.93 ^a^	12.24 ^ab^	9.40 ^b^	11.87 ^ab^	0.38	0.020	0.063	0.561
INS, mIU/L	48.98 ^b^	52.76 ^b^	60.76 ^b^	67.98 ^ab^	85.20 ^a^	3.08	<0.001	<0.001	0.149
PROG, pmol/L	1698.06	1660.56	1699.44	1696.67	1706.39	29.33	0.991	0.814	0.843
E2, pmol/L	111.13	106.16	112.07	108.31	117.31	2.32	0.643	0.396	0.367

SEM, standard error of the mean. Differences were assessed using Tukey’s test for multiple comparisons and denoted as follows: data within a row without a common letter are significantly different (*p* < 0.05).

**Table 4 animals-14-03044-t004:** Effects of different dietary net energy concentrations on plasma antioxidant capacity of sows.

Item	Dietary Net Energy Concentration, MJ/kg					SEM	*p*-Value		
	10.05	10.47	10.89	11.30	11.72		ANOVA	Linear	Quadratic
T-AOC, mM	0.42	0.40	0.39	0.40	0.39	0.03	0.998	0.823	0.889
T-SOD, U/mL	184.43 ^ab^	104.46 ^b^	118.13 ^b^	196.50 ^ab^	233.24 ^a^	13.76	0.005	0.024	0.004
MDA, nmol/mL	3.49 ^ab^	5.00 ^a^	4.09 ^ab^	3.81 ^ab^	1.63 ^b^	0.36	0.041	0.041	0.020

SEM, standard error of the mean. Differences were assessed using Tukey’s test for multiple comparisons and denoted as follows: data within a row without a common letter are significantly different (*p* < 0.05).

**Table 5 animals-14-03044-t005:** Effects of different dietary net energy concentrations on milk composition of sows.

Item	Dietary Net Energy Concentration, MJ/kg					SEM	*p*-Value		
	10.05	10.47	10.89	11.30	11.72		ANOVA	Linear	Quadratic
Milk fat content, %	6.78 ^ab^	6.12 ^b^	7.24 ^a^	7.35 ^a^	7.40 ^a^	0.14	0.007	0.005	0.667
Nonfat solids content, %	11.08	11.27	11.27	11.22	11.33	0.10	0.958	0.552	0.837
Lactose content, %	4.20	4.26	4.28	4.26	4.30	0.04	0.958	0.513	0.824
Lactoprotein content, %	6.04	6.13	6.13	6.10	6.15	0.05	0.976	0.641	0.794

SEM, standard error of the mean. Differences were assessed using Tukey’s test for multiple comparisons and denoted as follows: data within a row without a common letter are significantly different (*p* < 0.05).

**Table 6 animals-14-03044-t006:** Effects of different dietary net energy concentrations on immunoglobulins in plasma and milk of sows.

Item	Dietary Net Energy Concentration, MJ/kg					SEM	*p*-Value		
	10.05	10.47	10.89	11.30	11.72		ANOVA	Linear	Quadratic
Plasma									
IgA, μg/mL	32.49	33.83	35.17	30.73	32.74	0.79	0.508	0.651	0.515
IgG, μg/mL	190.00	190.44	197.46	180.13	184.74	3.75	0.688	0.454	0.625
IgM, μg/mL	36.81	34.37	39.85	40.05	45.15	1.25	0.066	0.010	0.314
Milk									
SIgA, μg/mL	25.05 ^b^	27.21 ^b^	27.00 ^b^	27.97 ^b^	34.53 ^a^	0.69	<0.001	<0.001	0.003
IgA, μg/mL	31.05 ^b^	31.12 ^b^	31.53 ^ab^	33.70 ^ab^	37.27 ^a^	0.74	0.021	0.003	0.111
IgG, μg/mL	365.32 ^b^	396.09 ^ab^	381.03 ^ab^	391.44 ^ab^	440.96 ^a^	7.81	0.020	0.005	0.273
IgM, μg/mL	34.99 ^b^	38.65 ^b^	39.40 ^b^	43.30 ^b^	47.74 ^a^	0.91	<0.001	<0.001	0.219

SEM, standard error of the mean. Differences were assessed using Tukey’s test for multiple comparisons and denoted as follows: data within a row without a common letter are significantly different (*p* < 0.05).

## Data Availability

All the datasets used and analyzed during the current study are included in the manuscript.

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
