# Peer review of "Effects of Dietary Net Energy Concentration on Reproductive Performance, Immune Function, Milk Composition, and Gut Microbiota in Primiparous Lactating Sows"

_animals, 2024, doi:10.3390/ani14203044_

Round 1

Reviewer 1 Report

Comments and Suggestions for Authors

Comments to the authors

Major comments

Material and methods

-        L96-97: you should provide more details about the experimental animals (e.g., age, mean BW, health status, vaccinations etc.)

-        L103: you should provide more detail about the litters (percentage of male and female piglets, use or not of replacer milk or creep feed)

-        L131-133: describe the method of BW and ADG estimation (per litter or piglet?).

-        L149-176: Add appropriate references.

Discussion 

-        You could include a paragraph of your thoughts about the significance of your results for the pig production.

Minor comments

-        You should improve issues of plagiarism (Percent match: 36%)

-        L18: … concentration based on the net…

-        L28: ….With the increasing dietary…

-        L220: .. There were no significant differences in the plasma 

-        L286: .. common letters indicate 

-        L302: .. common letter indicates..

-        L303: .. a highly significant..

-        L306: .. studies have shown..

-        L307: .. improved the energy intake of sows and the growth of suckling..

-        L330: .. In a previous study..

-        L331: .. the change in dietary energy…

-        L332: .. on the milk production …

-        L350: … immunoglobulins..

-        L356: A previous study found …

-        L370: .. impaired normal 

Reviewer 2 Report

Comments and Suggestions for Authors

1- write net energy (NE) in all references

2- lines 114-118: reference per kg of the diet. Consider reference per kg of premix. The Level of vitamin E is low, are you sure? I prefer reference vitamina us Vit A vs VA

3- Do you included fitases? Levels?

4- table 1 line 113: added  ME, MJ/kg plus NE. Refer  P total and Pdig

5- Do you consider measure lean content plus backfat at the same time or thinking this in new study?

6- I like a long discussion and consider a sort conclusion  

Reviewer 3 Report

Comments and Suggestions for Authors

This manuscript has investigated that the effect of net energy levels on sow reproductive performance in lactation. English needs to be improved. Discussion needs significant improvement before considering this for a publication.

L26 please revise to piglets, while

L28-30 plasma insulin levels, glucose levels, urea nitrogen levels.

L34 the milk immunoglobulin M, immunoglobulin G…

L52-53 please add a reference (s) for this sentence.

L57 what does ‘potential for sows to feed’ mean?

L65 ‘negative’ instead of ‘bad’

L68 Many previous studies

L77 Xue et al. (2012). L79 remove (Xue et al., 2012)

L80 Rooney et al. (2020). L82 remove (Rooney et al., 2020)

L96 please add initial backfat thickness across the treatments.

L99 the increment is not by 5%. It is increment of 100 kcal/kg from 10.05 MJ/kg. Please revise.

L121 please use gravity instead of rpm.

L123 what was the IU of oxytocin?

L137 Please add all assay kit’s catalog numbers. Please add dilution rate of each measurement. Please add how you processed the milk for analysis – centrifuged?

L181 Regression analysis is to obtain some equations for the results. For linear and quadratic effect statistics usually use orthogonal polynomial contrast.

L182 please declare p-value for significance and trend. Please add model used and experimental unit.

L183 Results: With linear and quadratic effects having a greater statistical power than ANOVA, the ANOVA p-values are not needed in the tables.

L184 please do not use a sentence in the subtitle in the Results.

Table 2 what is average of piglets? Is it piglet body weight?

L196-198 This sentence is not needed

L199 plasma GLU / BUN not plasma BUN as it is already blood urea nitrogen

L200-202 This sentence is not needed.

L203 plasma GLU has an effect. Please remove GLU.

L215-217 this sentence is not needed but the net energy level showing the highest level of MDA (10.47) and the lowest level of T-SOD (10.47) needs to be added.

L225-227  This sentence is not needed.

L204, L229, L244 No difference p-value should be p>0.10 but not p>0.05 as authors use p-value over 0.10 as an effect. In all results having 0.05<P<0.10, should have ‘tendency’ or ‘trend’ in the sentences.

L236-244 these sentences are not needed but SIgA is showing a quadratic effect so please revise it.

L310 please double check the format of citations throughout the manuscript.

L310 please add some details of the finding of Lu et al so that readers can compare.

L310 revise to ‘Although there was no significant difference in energy intake, the daily net energy intake of sows in the group of 11.72 MJ/kg increased by 5.73 MJ compared with the sows in the lowest energy group (10.05 MJ/kg).’

L314 What is Hazel? Rooney et al is correct citation? Please add more details about the results of the citation.

L314-315 reduce the body weight loss…

L314-315 What does this sentence mean?

L315-320 This part is not fit to discuss the results. Based on this study results, net energy intake may not change due to the dietary net energy concentrations. Does this mean that voluntary lactation feed intake is also controlled by energy needs similar with growing pigs reducing feed intake when they consume a high energy diet but their energy intake would not change? Authors need to discuss this. Also, if authors would like to declare the numerical difference of 5.73 MJ daily energy intake, there should be more discussion about why increased energy intake (although it was numerically) did not affect sow and piglet body weight change/growth.

L325 S. et al is incorrect citation. Also L323-325 sentence needs to be revised as the cited study only used multiparous sows.

L326-328 I agree with this statement but further discussion is needed for the results in the current study. As this study used primiparous sows, is no difference in piglet growth due to limited milk production? But this study showed increased milk fat content and immunoglobulins in milk.

L338-339 incomplete sentence?

L342 please cite those ‘studies’

L354-355 Are there no study showing increased Ig by increasing net energy concentrations? Further discussion is needed.

L365 Glucose level increased linearly with increasing net energy concentrations with a tendency.

L367 Further information is needed for how increased plasma insulin levels by increased energy level doesn’t mean negative effect in insulin regulation. Also need further discussion regarding how increased dietary energy by fat could increase insulin level, how no difference in energy intake but in feed intake affected blood glucose and insulin levels.

Also the sentence in L366-367 needs to be re-written.

L373-374 Further discussion is needed. How energy level or soybean oil(?) could affect antioxidant capacity and oxidative stress. Also, the results showed quadratic responses where 10.47 MJ/kg treatment had the lowest SOD and highest MDA. What happens in 10.05 (lower) energy treatment?

L382 Citation is different between the beginning and end of the sentence.

L398 I disagree with this sentence. This effect may be confirmed if piglet blood Ig concentrations as milk Ig levels increased.

Comments on the Quality of English Language

There are several sentences that don't read well and are incomplete.

Also sentences are not well flowed for discussion.

Round 2

Reviewer 3 Report

Comments and Suggestions for Authors

Comments 5: L57 what does ‘potential for sows to feed’ mean?

Response 5: Thank you for your inquiry. “potential for sows to feed” means the maximum feed intake that a sow can theoretically achieve.

è Please revise this by adding more details.

Comments 10: L96 please add initial backfat thickness across the treatments.

Response 10: Thank you for pointing this out. In response to your suggestion, we have added the initial backfat thickness across the treatments.

è What about body weight? Body weight and its change are also important measurements. Please add this data.

Comments 13: L123 what was the IU of oxytocin?

Response 13: Thank you for your inquiry. The concentration of oxytocin was 1ml:10 IU.

è Please add IU to the text. So 20 IU?

Comments 14: L137 Please add all assay kit’s catalog numbers. Please add dilution rate of each measurement. Please add how you processed the milk for analysis – centrifuged?

Response 14: Thank you for pointing this out. We have added all assay kit’s catalog numbers, dilution rate and the the processing method of milk.

è Temperature for milk centrifugation needs to be added. Blood as well (L121)

Comments 16: L182 please declare p-value for significance and trend. Please add model used and experimental unit.

Response 16: Thank you for pointing this out. We have declared p-value for significance and trend, and added the model used and experimental unit.

è Please add model term for fixed and random effect – treatment – fixed and rep – random?

Comments 17: L183 Results: With linear and quadratic effects having a greater statistical power than ANOVA, the ANOVA p-values are not needed in the tables.

Response 17: Thank you for your suggestions. Because the ANOVA p-values can help to fully present the statistical information in the model, retaining these p-values can increase the transparency of the research results and provide a more thorough understanding of the statistical significance. Therefore, we decided to keep the p-values.

è Please add superscripts for results having tendency in all tables.

Comments 29: L310 please add some details of the finding of Lu et al so that readers can compare.

Response 29: Thank you for pointing this out. In response to your suggestion, we have added the sentence.

The added sentence is as follows:

“Their study found that increasing dietary net energy concentration from 2.11 Mcal/kg to 2.73 Mcal/kg resulted in a linear decrease in average daily feed intake of barrows and gilts. Interestingly, in their study, comparing to pigs fed 2.42 Mcal/kg NE diet, even though those fed 2.11 Mcal/kg NE diet had 8.7% increment on ADFI, daily NE intake was still 5.4% less. This is consistent with our results.”

è Please use same unit for energy levels throughout the study. So convert Mcal to MJ.

Comments 32: L314-315 reduce the body weight loss…

Response 32: Thank you for pointing this out. In response to your suggestion, we have revised the sentence.

è I couldn’t find where the revised sentence is

Comments 33: L314-315 What does this sentence mean?

Response 33: Thank you for your inquiry. This sentence means that inadequate feed intake affects the level of nutrient intake of the sow, which in turn affects the sow's weight loss.

è I couldn’t find where the revised sentence is

è Please add this detail in the text.

Comments 34: L315-320

Response 34: “In this study, it was noted that an increase in net energy intake of 5.73 MJ per day did not have a significant effect on backfat loss in primiparous sows. This may be because body fat reserves are the most backward in the energy allocation order of lactating sows, and a bit increase of the daily energy intake cannot affect the body storage of sows. Previous studies also found that an increase of 8.26MJ ME in daily energy intake had no significant effect on weight loss in primiparous sows.”

è L332-336. This part needs references. Please don’t use ‘a bit’. I disagree that 100 kcal increase per each treatment is not small increase.

Comments 36: L326-328 I agree with this statement but further discussion is needed for the results in the current study. As this study used primiparous sows, is no difference in piglet growth due to limited milk production? But this study showed increased milk fat content and immunoglobulins in milk.

Response 36: Thank you for your inquiry and suggestions. We found that milk yield and amino acid levels may be the main factors affecting the growth performance of piglets.Primiparous sows were used in this study, and limited milk production is also one of the main factors affecting the growth performance of piglets. We will also focus on the above factors in the follow-up study.

è I couldn’t find where the revised sentence is

Comments 37: L338-339 incomplete sentence?

Response 37: Thank you for pointing this out. In response to your suggestion, we have revised the sentence.

è I couldn’t find where the revised sentence is

Comments 38: L342 please cite those ‘studies’

Response 38: Thank you for your suggestions, we have added the citations of those studies.

è I couldn’t find where the revised sentence is

L337 Please revise to ‘Interestingly, in this trial, although the daily energy intake of sows was not statistically different among dietary treatments, there were significant…

Comments 45: L398 I disagree with this sentence. This effect may be confirmed if piglet blood Ig concentrations as milk Ig levels increased.

Response 45: Thank you for pointing this out. In response to your suggestion, we have revised this sentence.

è I couldn’t find where the revised sentence is

Comments on the Quality of English Language

The flow of discussion may need some attention.
